# Some Structural Elements of Bacterial Protein MF3 That Influence Its Ability to Induce Plant Resistance to Fungi, Viruses, and Other Plant Pathogens

**DOI:** 10.3390/ijms242216374

**Published:** 2023-11-15

**Authors:** Denis Erokhin, Sophya Popletaeva, Igor Sinelnikov, Alexandra Rozhkova, Larisa Shcherbakova, Vitaly Dzhavakhiya

**Affiliations:** 1All-Russian Research Institute of Phytopathology, 143050 Bolshie Vyazemy, Russia; erokhin.denis.v@gmail.com (D.E.); unavil@yandex.ru (S.P.); dzhavakhiya@yahoo.com (V.D.); 2Federal Research Centre “Fundamentals of Biotechnology”, Russian Academy of Sciences, 119991 Moscow, Russia; i.sinelnikov@fbras.ru (I.S.); amrojkova@yahoo.com (A.R.)

**Keywords:** structure–function relationship, plant resistance, MF3 inducer, 3D models

## Abstract

The ability of the MF3 protein from *Pseudomonas fluorescens* to protect plants by inducing their resistance to pathogenic fungi, bacteria, and viruses is well confirmed both in greenhouses and in the field; however, the molecular basis of this phenomenon remains unexplored. To find a relationship between the primary (and spatial) structure of the protein and its target activity, we analyzed the inducing activity of a set of mutants generated by alanine scanning and an alpha-helix deletion (ahD) in the part of the MF3 molecule previously identified by our group as a 29-amino-acid peptide working as the inducer on its own. Testing the mutants’ inducing activity using the “tobacco–tobacco mosaic virus” pathosystem revealed that some of them showed an almost threefold (V60A and V62A) or twofold (G51A, L58A, ahD) reduction in inducing activity compared to the wild-type MF3 type. Interestingly, these mutations demonstrated close proximity in the homology model, probably contributing to MF3 reception in a host plant.

## 1. Introduction

The global population is expected to reach 10 billion people by the middle of the 21st century, and this growth will be accompanied by a permanent rise in food consumption [1]. To produce more food, farmers will have to accelerate the use of chemical pesticides to effectively control crop losses caused by plant diseases. However, the increased application of pesticides leads to issues with pests’ resistance and pollution caused by pesticide residues [2]. Thus, alternative approaches that make it possible to avoid or significantly reduce the use of pesticides may be employed to address their overuse.

Such approaches include a number of eco-based methods. For example, organic fertilization provides restoration of biodiversity of soil fungi, thus reducing the level of plant infection by pathogenic fungi [3]. The use of antagonistic fungi, yeasts, and bacteria provides some control of plant pathogenic fungi [4,5,6], while the combination of some antagonistic fungi or some compounds of natural origin with commercial fungicides makes it possible to significantly reduce the dosages of these chemicals, thus reducing the pesticide load on the environment [7,8].

Another group of methods is based on plant immunity improvement achieved by the stimulation of plants with electromagnetic fields [9] or the priming of plants with nonpathogenic microorganisms [10] as well as microbial or plant metabolites. Application of inducers for activation or priming of plant resistance to various pathogens is a strategy that relies on nonspecific plant defense responses, promoting more sustainable agriculture [11,12]. Inducers can be substances of different natures and origins. In many cases, inducers trigger one or both of the mainstream pathways leading to the development of nonspecific resistance in the entire plant: systemic acquired resistance (SAR) [13] and induced systemic resistance (ISR) [14].

Typical synthetic inducers of systemic resistance are analogues of some plant hormones. Most of them belong to the family of salicylic acid (SA) derivatives and its structural and/or functional analogues [15,16,17]. They are involved in SAR. Besides the SA derivative methyl salicylate, other functional analogues of SA such as glycerol-3 phosphate, dehydroabietinal, and azelaic acid can act as mobile signals of SAR [18]. Jasmonic acid and ethylene are the key plant hormones associated with ISR [19]; however, in the last decade, experiments showed some evidence that salicylic acid is also involved in this mechanism [14]. There are also a number of nonhormone-like inducers including vitamins [20,21,22], nonprotein amino acids [23,24], and even inorganic salts [25,26].

Most of the currently known biogenic inducers are microbial metabolites. They are produced by fungi, bacteria, viruses, or oomycetes and act as inducers of plant immunity when interacting with the host plant. They may have variable structures, the most common of which are oligosaccharides, peptides, glycopeptides, and proteins [11]. The following examples of well-studied biogenic inducers represent their main structural classes: β-glucans [27,28,29], chitin [30,31], and chitosan [32,33] are the typical fungal oligosaccharide inducers; flg22 [34,35,36,37,38] is a peptide inducer from bacterial flagella; BcGs1 [39] is a fungal glycopeptide inducer isolated from *Botrytis cinerea*; and bacterial harpins [40,41,42] are examples of protein inducers.

Although many synthetic inducers demonstrate high protective activity against crop diseases [11], their use in agriculture should be considered with caution, and, as with other xenobiotics, they should be carefully checked for phytotoxicity and biodegradability. Contrary to that, being ubiquitous in biomes, biogenic inducers may put less pressure on the environment because of their higher rate of biodegradation compared to xenobiotics and do not result in the accumulation of toxic residues. Moreover, in providing defense to plants, some biogenic inducers may also be beneficial for plant health. For example, harpin Hpa1 stimulates growth in rice with little cost to plant development [43,44].

The special interest in bacterial protein inducers is caused by the added benefit of leveraging powerful and flexible recombinant protein technology [45] to produce active ingredients for commercial use. One such inducer, a low-molecular-weight (16.9 kDa), thermostable protein named microbial factor 3 (MF3), was isolated from the plant-growth-promoting rhizobacterium *Pseudomonas fluorescens* [46]. MF3 ability to protect plants both in greenhouses and in the field by inducing resistance to some major groups of plant pathogens including fungi [46,47] and viruses [46] was confirmed. The protective properties of MF3 were primarily revealed in dicot plants in the following host–pathogen pairs: tobacco—*Alternaria longipes*, tobacco–tobacco mosaic virus (TMV), and white cabbage (*Brassica oleracea* var. *capitata*)–turnip mosaic virus [46]. Interestingly, MF3 did not directly affect TMV or *A. longipes*, and it also possessed no phytotoxicity. Further experiments with monocot plants confirmed the protective activity of MF3 in wheat—*Septoria nodorum* and rice—*Magnaporthe oryzae* (*syn. Pyricularia oryzae*) pathosystems [47]. Preliminary experiments also indicate the presence of a protective effect of MF3 in relation to some plant-infecting bacteria and oomycetes. After the identification of the MF3 gene structure, it was found that MF3 belongs to a family of FKBP-type peptidyl-prolyl *cis–trans* isomerases [46].

We supposed that the active center of MF3 (AC MF3) responsible for its ability to induce plant resistance is localized within the most conserved sequence of this protein. The calculation of this sequence using a PROSITE bioinformational resource showed this sequence consisted of 29 amino acid residues (Figure 1) and contained an arginine−lysine site that could be subjected to a trypsinolysis resulting in a complete lack of inducing activity of MF3; therefore, AC MF3 is suggested to be a mandatory part of the entire inducer [48]. We also showed that the synthetic AC MF3 peptide protected by bovine serum albumin from protease degradation possessed its own inducing activity comparable with that of the wild-type MF3.

Despite some progress in AC MF3 isolation and description, the molecular basis of its inducing activity and inducing activity of the entire MF3 remain poorly explored. The revealing of the relationship between the structural elements of the inducer and their influence on its inducing activity [49,50,51] is valuable on its own but also may become the first step towards identification of a relevant plant receptor, since the first molecular event that normally triggers inducer–plant interaction is the inducer recognition by receptors of plant cells [52].

The current study was aimed at the further molecular characterization of MF3 and its active center by investigating relationships between some elements of the primary and predicted secondary structures of AC MF3 and the inducing activity of the entire protein. For this purpose, a site-specific mutagenesis was realized within AC MF3, including a number of point mutations and a removal of the predicted alpha-helix fragment. The obtained mutants were tested for their inducing activity using a tobacco–TMV pathosystem. To determine the spatial distribution of the studied mutations, the tertiary MF3 structure was modeled. The results of this study may narrow the functional region of AC MF3 and, considering this region to be a putative receptor-binding site, they may be also helpful in its further in silico docking to plant receptors with the resolved 3D structures. Protein mutants associated with the maximum reduction of the inducing activity of MF3 may be further used as discriminators in affine chromatography experiments aimed at the identification of the corresponding receptors in plant cells.

## 2. Results

### 2.1. Design of AC MF3 Mutants

A prediction of the secondary structure of AC MF3 was carried out using both a computation method (Figure 2a) and multiple alignment of the target sequence with sequences with the known 2D structures (Figure 2b). The resulting combined secondary structure of AC MF3 is shown in Figure 2c. Eleven mutants including some point substitutions to alanine and deletion of the predicted alpha-helix region were designed (Figure 2d) and obtained.

### 2.2. Inducing Activity of AC MF3 Mutants

To reveal mutations influencing the MF3 protective properties, ten mutants of this protein with single amino acid substitutions (G51A, D57A, L58A, V60A, V62A, P64A, E65A, D66A, Y68A, G69A) generated by alanine scanning and one mutant with the alpha-helix deletion (ahD) were produced via heterologous expression. The ability of the obtained mutants to induce plant resistance was tested in parallel with wild-type MF3 using a tobacco–TMV pathosystem. In these experiments, the target activity of all obtained mutants except G69A turned out to be lower than that of MF3 (Figure 3).

Among the studied mutants, V62A (valine substitution with alanine at position 62) was surprisingly the least active. A pre-inoculation application of V62A on tobacco leaves resulted in an almost threefold reduction of the number of developed necrotic lesions compared to wild-type MF3. Other mutants were divided into three groups according to their activity level. The first group included G51A, V60A, and L58A containing alanine-substituted glycine, valine, and leucine, respectively, as well as ahD; these mutants showed approximately twice lower target activity than wild-type MF3 (see Figure 4 as an illustrative example). Mutants from the second group (P64A and D66A) provided a 1.5-fold reduction of the number of developing necrotic lesions, while the third group (D57A, E65A, and Y68A) provided statistically significant (1.2–1.3-fold), but rather biologically unimportant suppression of the disease manifestation.

### 2.3. The 3D Model of MF3 and Its Analysis

A 3D model of MF3 was designed using a homology approach (Figure 5). The model showed a high confidence level (0.65–0.83) for the spatial structure of AC MF3.

A spatial distribution of amino acids, whose mutations led to the maximum reduction of the inducing activity of MF3, is shown in Figure 6.

The performed analysis revealed that Gly 51, Leu 58, Val 60, Val 62, and the alpha-helix, which met the above-mentioned criterion, are located close to each other on the same side of the MF3 globule.

## 3. Discussion

Identification of the relationship between the structure of the protein under study and its function is one of the most important tasks in the molecular characterization of proteins. Such research may shed light on which elements of the primary, secondary, or tertiary structure of a protein are responsible for the manifestation of its functional activity. Earlier we showed that the 29-amino-acid region (AC MF3) of the MF3 molecule is obligatory for its inducing activity [48], although it remained unclear which specific elements of AC MF3 were associated with the manifestation of this activity. We designed and obtained 11 mutant proteins characterized by some point substitutions to alanine or a deletion of the predicted alpha-helix (ahD mutant) within AC MF3. The method of amino acid substitutions to alanine (alanine scanning approach [58]) was chosen because it helps to avoid any bias in the choice of the amino acid to be replaced. Alpha-helixes are known as common sites for the protein–receptor interaction [59]. Thus, the purpose of obtaining this ahD mutant was to test whether this particular alpha-helix is important for the inducing activity of MF3. In addition, we hypothesized that the deletion of this alpha-helix would result in a greater reduction of the target MF3 activity compared to point mutations, as it was suggested to more significantly interfere with the structure of AC MF3. The choice of the tobacco (var. Xanthi)–TMV assay for the validation of the inducing activity of MF3 was determined by its sensitivity and a large amount of necroses usually developing on the leaves of this tobacco variety, allowing the quantification of the results of the experiment. A 3D model of MF3 was developed to analyze the spatial distribution of the studied mutations. A homology method [55] was used for modeling since it relies on experimental data rather than on computational algorithms only. As a template for modeling, a SlyD PPI-ase (8g01.1.F) from *Esherichia coli* was chosen, since it represents the most relevant MF3 homologue with resolved 3D structure.

The results of the study of the inducing activity of MF3 mutants were rather unexpected for us. First, we did not expect that the maximum effect on the inducing activity of MF3 would be shown by the V62A mutation representing the alanine substitution of a hydrophobic valine, rather than by the substitution of a polar or charged amino acid or proline known to affect the rotation of polypeptide chains. Second, though the alpha-helix deletion resulted in a notable reduction of the target MF3 activity, this reduction was comparable to that for a glycine substitution and even weaker than the effect of a valine substitution. This result can be explained by the suggestion that the target activity of the protein is probably influenced not by the whole alpha-helix but rather some individual amino acid or amino acids within it. Or, perhaps, the answer is that AC MF3 is not the only active center in the MF3 molecule responsible for its target activity, and, for example, a plant cell receptor may bind not only to AC MF3. It is worthwhile to perform further alanine scanning experiments with the other amino acids composing AC MF3, including those within the alpha-helix, in order to refine the data on the contribution of individual amino acids to the plant resistance-inducing activity of MF3.

The obtained 3D MF3 model showed that the structure of AC MF3 has a high level of confidence. Thus, it can be used to make assumptions about the spatial distribution of the mutations of interest. Those amino acids (Gly 51, Leu 58, Val 60, Val 62) and alpha-helix, whose substitution or deletion led to a 2–2.5-fold decrease in the target MF3 activity, demonstrate proximity to each other and form a compact region on the surface of the protein globule, while amino acids associated with a reduction of 1.5-fold or less in the target MF3 activity are located away from this site (Figure 6). This may be interpreted as the presence of a putative receptor-binding site within the AC region of MF3. Combining data on the plant resistance-inducing activity of mutant proteins and the spatial distribution of the corresponding amino acids, one can assume that the functional domain of AC MF3, which is probably responsible for binding with a cell receptor, may be narrowed from 29 to about 15 amino acids. Note, however, that this assumption can be clarified only after the completion of more alanine scanning experiments.

The obtained data can be applied for the further molecular characterization of MF3. First, the putative receptor-binding site within AC MF3 can be used for in silico docking to plant receptors with known structures. Second, protein variants with mutations providing the maximum negative influence on the inducing activity of MF3 may be used as negative discriminators in affine chromatography experiments aimed at the identification of putative plant receptors.

Elucidation of the role of amino acids composing AC MF3 may be of both theoretical and practical significance. Such information may help to identify a putative corresponding plant receptor that, in turn, will provide a better understanding of the mechanisms of the inducing activity of MF3. At the same time, this information can be used to design an improved version of MF3 or its active center, which would be characterized by a higher activity and probably consist of fewer amino acids, thus requiring less effort for its biosynthesis and stability maintenance. The results of laboratory studies [46,60,61] and some data obtained in completed [62] and still ongoing greenhouse and field trials with such crops as potato, tomato, and wheat evidence a good protecting efficiency of MF3 against *Phytophthora infestans* and *Stagonospora nodorum* as well as potato and tomato viruses. The development of an improved (more efficient or more stable) version of this plant resistance inducer may accelerate the development of corresponding environmentally safe biopreparations intended to improve resistance of agricultural crops to various pathogens. Note that no chemical pesticides have been registered for direct use against plant viruses yet, though this type of plant pathogen is able to cause serious yield losses [63], especially in the case of a combination of several viruses or viruses with fungal pathogens [64]. The control of viral diseases in crops includes mainly risk-reducing measures, such as the use of virus-resistant crops and insecticides to kill carriers (the majority of plant viruses are carried by insects) and quality control of seed material to prevent the further spreading of infection [65]. To date, the main way to deal with virus-infected plants is to eradicate them to prevent the further spreading of infection. In view of these difficulties related to the nature of viral pathogens, the appearance of preparations able to directly reduce the level of viral (as well as fungal) infections in crops may become a new step in providing food safety and improving yield quality.

## 4. Materials and Methods

### 4.1. Design and Obtaining of AC MF3 and Alpha-Helix Deletion Mutants

A QuikChange^TM^ [66] method was chosen to obtain mutations in AC MF3 designed by the alanine scanning procedure [58]. QuikChange^TM^ uses a pair of partially complementary oligonucleotides with an introduced mutation. During PCR, oligonucleotides were annealed on matrix DNA representing the starting pET28-MF3 plasmid. Plasmid DNA containing the target mutation was further amplified. To avoid the occurrence of nontarget mutations, a high-precision Q5^TM^ DNA polymerase (NEB, Ipswich, MA, USA) was used for amplification.

The amplification mode used in this study included 15 cycles at 98 °C for 10 s, 60 °C for 30 s, and 72 °C for 300 s. The linear PCR product represented a linearized vector with overlapping complementary ends. The resulting pool of DNA (original and mutant) was then treated with DpnI restriction endonuclease to degrade the original DNA. Linear mutant DNA obtained after such treatment was then transformed into *E. coli* NEB Turbo cells, where the break was further ligated along the homology site by a bacterial reparation system with the introduction of mutations embedded in primers. The primers used for mutagenesis are presented in Table 1.

The alpha-helix deletion was obtained by a similar method, using forward primers with two homology sites of 19 nucleotides complementary to the original sequence and a deletion between them.

As a result, 11 plasmids containing AC MF3 mutants were obtained. The plasmids were transformed into the expression strain (*E. coli* Rosetta (DE3)), and the homogeneous forms of 11 mutant proteins were isolated using Ni Sepharose immobilized metal ion affinity chromatography (IMAC) [67].

The results of a Sanger sequencing of the AC MF3 region containing the ahD deletion are shown in Figure 7.

### 4.2. Inducing Activity Assay

To assess the resistance-inducing activity of the obtained mutants compared to wild-type MF3 and to examine the relationship between this activity and the protein structure, a bioassay involving inoculation of detached leaves of necrotic-responding tobacco cultivar Xanthi NN with TMV was used (see [60] for details). In brief, five 10 µL drops of a tested mutant protein solution were applied on the left half of each detached leaf and spread on its adaxial surface (10 leaves per treatment). The right half of the same leaf was similarly treated with a solution of the wild-type MF3 protein. Protein solutions were prepared on the day of the experiment by dissolving freeze-dried preparations of MF3 or mutant proteins purified by IMAC (see Section 4.1) in distilled water. Prior to the experiments, the concentrations of mutant proteins were adjusted to that of MF3 using a Qubit 4 fluorometer (Thermo Fisher Scientific, Waltham, MA, USA). Treated leaves were incubated in a wet chamber for 24 h followed by a separate inoculation of left and right halves with equal concentrations of TMV by rubbing them with a mixture of a viral suspension and carborundum [46]. MF3 was used at a concentration of 10 µg/mL, which was determined earlier to result in a 50–70% reduction of the number of necrotic lesions compared to a water-treated control. After a 4–5-day incubation of inoculated leaves in a wet chamber at 20–22 °C, the level of inducing activity of each mutant was evaluated by a comparison of the number of necrotic lesions developed on a leaf half treated with a mutant with that obtained for the right leaf half treated with MF3; the result was expressed as a percentage of increase or reduction of this index. The number of necroses developed after a TMV infection of tobacco plants represents a common index used for evaluation of the disease development level [68,69].

### 4.3. The 2D Structure Prediction and 3D Modeling

A 2D structure prediction was performed using a JPred server [53] with a JNET algorithm [54] and SWISS-MODEL [55]. A 3D modeling was carried out using SWISS-MODEL with a GMQE score for choosing the most relevant template. The model visualization and the further analysis were carried out using RCSB PDB [56] and Mol* 3D Viewer [57] programs.

### 4.4. Statistical Analysis

For each mutant, evaluation of its inducing activity arranged in parallel with the wild-type MF3 testing included three independent experimental series. Mean values, standard errors (SEs), and the significance of differences at *p* ≤ 0.05 (*t*-test for independent variables) were calculated using STATISTICA v. 6.1 software (StatSoft Inc., Tulsa, OK, USA).

## 5. Conclusions

To study the relationship between the primary structure of the MF3 protein, its spatial configuration, and ability to induce plant resistance to pathogens, a set of 11 mutants generated by alanine scanning and alpha-helix deletion (ahD) in the earlier revealed 29-amino-acid region (AC MF3) responsible for the target activity were generated and tested for their protecting activity in a tobacco–TMV model system. Several mutants showed almost threefold (V60A and V62A) or twofold (G51A, L58A, ahD) reduction in their inducing activity compared to the wild-type MF3. A construction of the 3D MF3 model showed these mutations were located in close proximity to each other within AC that may be interpreted as the presence of a putative receptor-binding site in this region of the MF3 molecule. The data obtained can be applied for the further molecular characterization of MF3 and identification of a putative plant receptor for this protein, while further alanine scanning experiments with other amino acids composing AC MF3 will refine the data on the contribution of individual amino acids to the inducing activity of this protein.

## Figures and Tables

**Figure 1 ijms-24-16374-f001:**
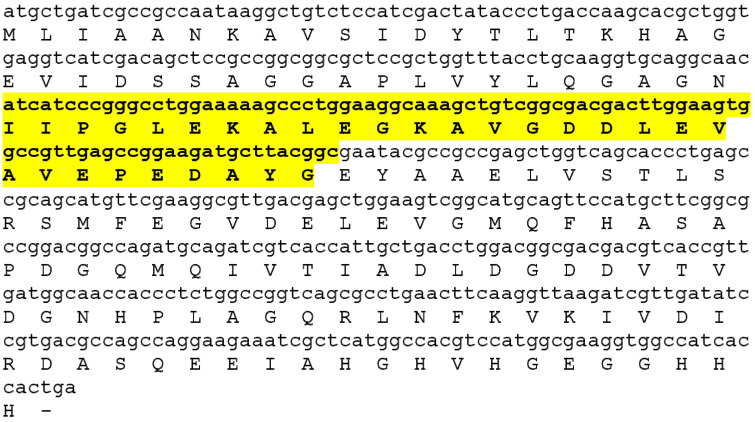
MF3-encoding gene and amino acid sequence. The MF3 active center is marked in bold and highlighted with yellow.

**Figure 2 ijms-24-16374-f002:**
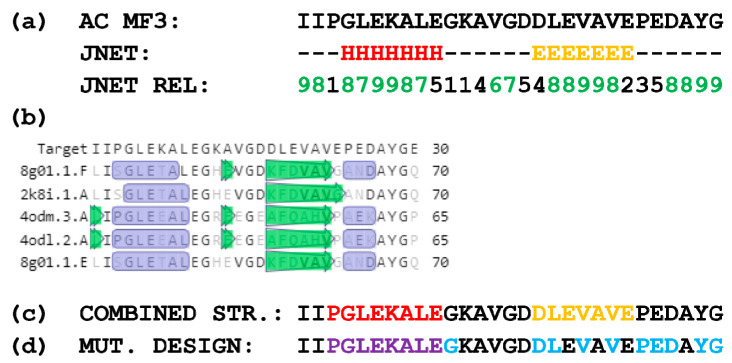
Prediction of the AC MF3 secondary structure and design of mutants. (**a**) Predicted secondary structure of AC MF3 calculated using a JPred server [53] and a JNET algorithm [54]: “H” indicates the alpha-helix region, “E” indicates a beta-strand; numbers represent the confidence level increasing from 1 to 9. (**b**) Predicted secondary structure of AC MF3 obtained using SWISS-MODEL [55]: purple boxes indicate alpha-helixes, green arrows indicate beta-strands. (**c**) Combined secondary structure: the alpha-helix and beta-strand are indicated with red and orange colors, respectively. (**d**) Mutation design: point substitutions to alanine are indicated with blue; the deleted alpha-helix motif is indicated with purple.

**Figure 3 ijms-24-16374-f003:**
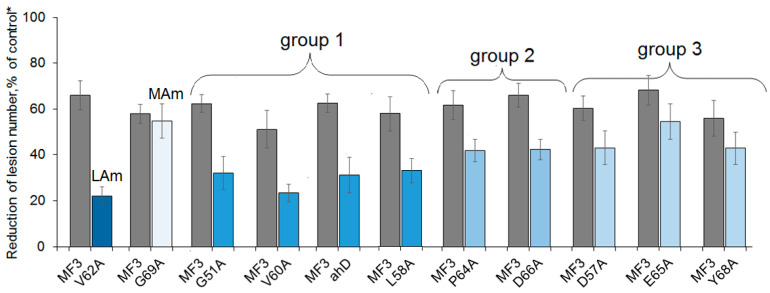
Comparison of the effect of the MF3 protein and its mutants on the formation of necrotic lesions on tobacco leaves artificially inoculated with TMV. LAm, the least active mutant; MAm, the most active one. Y-bars indicate standard error (SE) of the mean at *p* ≤ 0.05 (10 leaves per treatment, 3 replications per experiment). * Leaves treated with distilled water prior to inoculation were used as the control.

**Figure 4 ijms-24-16374-f004:**
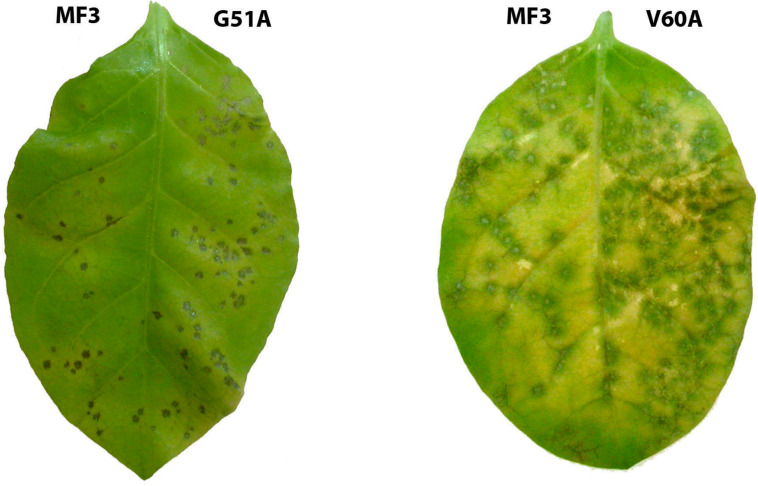
Examples of necrotic lesions developed on tobacco leaves, which were separately treated with wild-type MF3 (left halves) and G51A or V60A mutants (right halves) prior to inoculation with tobacco mosaic virus.

**Figure 5 ijms-24-16374-f005:**
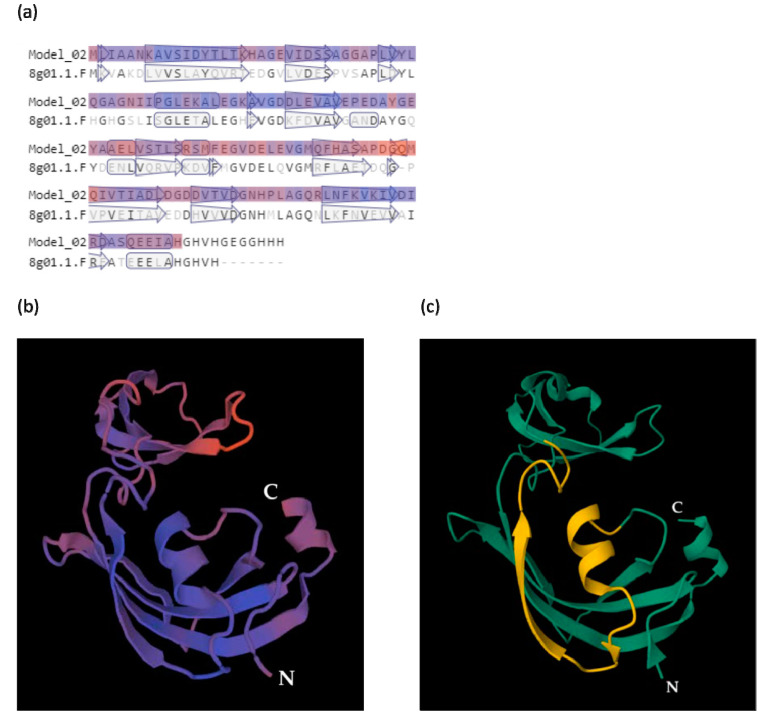
A 3D model of MF3 built with SWISS-MODEL [55] (alpha-helix regions are indicated with arrows). (**a**) Sequence alignment with the template 8g01.1.F. Here and in (**b**): more or less confident 3D structures are indicated with blue or red, respectively. (**b**) A 3D model of MF3, 2D structure representation. (**c**) AC MF3 region (indicated in yellow) visualized using RCSB PDB [56] and Mol* 3D Viewer [57]. N and C indicate the N- and C-end of the MF3 molecule, respectively.

**Figure 6 ijms-24-16374-f006:**
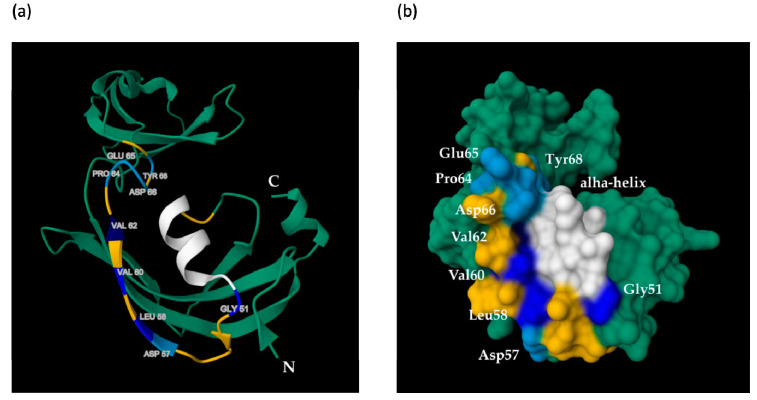
Spatial distribution of substituted amino acids within the MF3 molecule modeled using RCSB PDB and Mol* 3D Viewer: (**a**) 2D structure representation, (**b**) molecular surface representation. The alpha helix is marked with a white color. Amino acids associated with a <1.5-, 2-, and >2.5-fold reduction of the inducing activity of a protein are indicated with light-blue, blue, and dark-blue (Val62) colors, respectively. The rest part of the AC MF3 region is indicated with yellow. The rest of the protein molecule is indicated with green.

**Figure 7 ijms-24-16374-f007:**
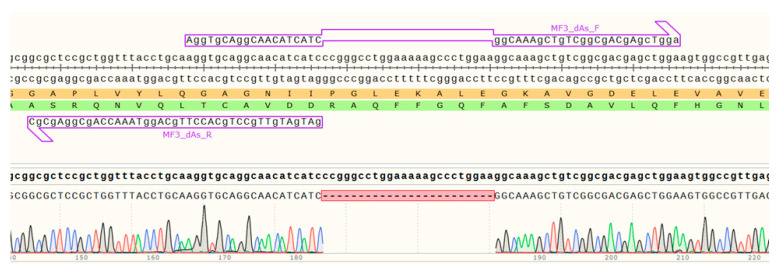
The target sequence of the MF3 region of the ahD mutant protein confirmed by Sanger sequencing of the plasmid used for its production. Amino acid sequences corresponding to the direct and complementary DNA sequences of a plasmid are indicated with yellow and green, respectively. Red indicates the alpha-helix deletion.

**Table 1 ijms-24-16374-t001:** Primers used for the AC MF3 mutagenesis.

No.	Primer	5′–3′ Sequence
1	2	3
1	11(G_A)F	cctggaagccaaagctgtcggcg
2	11(G_A)R	ctttggcttccagggctttttc
3	17(D_A)F	tcggcgacgccttggaagtggc
4	17(D_A)R	acttccaaggcgtcgccgacag
5	18(L_A)F	gcgacgacgcggaagtggccgttg
6	18(L_A)R	ccacttccgcgtcgtcgccgacag
7	20(V_A)F	ttggaagcggccgttgagcc
8	20(V_A)R	aacggccgcttccaagtcgtc
9	22(V_A)F	gtggccgctgagccggaagatg
10	22(V_A)R	tccggctcagcggccacttccaag
11	24(P_A)F	gttgaggcggaagatgcttacg
12	24(P_A)R	tcttccgcctcaacggccacttc
13	25(E_A)F	ttgagccggcagatgcttacggc
14	25(E_A)R	aagcatctgccggctcaacggcc
15	26(D_A)F	cggaagctgcttacggcgaatac
16	26(D_A)R	gtaagcagcttccggctcaacgg
17	28(Y_L)R	gatgctttaggcgaatacgccg
18	28(Y_L)F	attcgcctaaagcatcttccgg
19	29(G_A)F	tgcttacgccgaatacgccgcc
20	29(G_A)R	cgtattcggcgtaagcatcttcc
21	MF3_dAs_F	AggTgCAggCAACATCATCggCAAAgCTgTCggCgACgAgcTgga
22	MF3_dAs_R	gATgATgTTgCCTgCACCTTgCAggTAAACCAgCggAgCgC

## Data Availability

The authors declare that the data supporting the findings of this study are available within the main text of the manuscript. Raw data are available from the corresponding author upon reasonable request.

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
