# Peer review of "Some Structural Elements of Bacterial Protein MF3 That Influence Its Ability to Induce Plant Resistance to Fungi, Viruses, and Other Plant Pathogens"

_ijms, 2023, doi:10.3390/ijms242216374_

Round 1
Reviewer 1 Report
Comments and Suggestions for Authors
In this manuscript, the authors analyzed the inducing activity of a set of mutants in Pseudomonas fluorescens MF3 protein and found some sites that play an important role in the MF3 activity inducing activity using the tobacco–Tobacco Mosaic Virus pathosystem.
Although 2- and 3-fold reductions in inducing activity were found in the mutant proteins compared to the wild MF3 type, no further experiments were performed to show the mechanism, and no receptors in the plant were found associated with the induction. Therefore, I would not suggest to publish this manuscript in IJMS at this stage.
Moreover, still some points need to be addressed as follows:
1. It would be better to show the typical tobacco leaves together with Figure 3.
2. The authors use the percent reduction of the lesion number on each leaf half to evaluate the reduction of inducing activity, however, the area of each lesion was different, so it is proper to use the area of all the lesions on each treatment.
3. “2.1. Analysis of AC MF3 mutants inducing activity” here the subtitle should be 2.2; and “2.2. 3-D Model of MF3 and its analysis” here should be 2.3.
Author Response
Although 2- and 3-fold reductions in inducing activity were found in the mutant proteins compared to the wild MF3 type, no further experiments were performed to show the mechanism, and no receptors in the plant were found associated with the induction. Therefore, I would not suggest to publish this manuscript in IJMS at this stage.
We thank a reviewer for valuable comments.
The aim of our study was to investigate the relationship between some structural elements of MF3 and their functional activity. A similar approach focused mainly on studying the relationship between elicitor’s structure and function has been previously used by other authors (1-3) and, in our humble opinion, can work on its own. Please, note that at this stage the uncovering the mechanisms of action of MF3 or identifying the relevant plant receptors was beyond the scope of this particular study. To deal with these two questions, we are currently working on studying the response of a tobacco transcriptome to MF3. The results of this study will be presented in a separate manuscript. Taking into account your comments and in order to avoid some misunderstanding, we have changed the title of the manuscript and made some corrections in the main text, making it more specific and focused on the structure-function relationships. We also added the below-listed references to support our experimental design. All changes in the text are highlighted in yellow.
- Kunze, G.; Zipfel, C.; Robatzek, S.; Niehaus, K.; Boller, T.; Felix, G. The N terminus of bacterial elongation factor Tu elicits innate immunity in Arabidopsis plants. Plant Cell 2004, 16(12), 3496–3507. https://doi.org/10.1105/tpc.104.026765
- Noda, J.; Brito, N.; González, C. The Botrytis cinerea xylanase Xyn11A contributes to virulence with its necrotizing activity, not with its catalytic activity. BMC Plant Biol. 2010, 10, 38. https://doi.org/10.1186/1471-2229-10-38.
- Frías, M.; González, M.; González, C.; Brito, N. A 25-residue peptide from Botrytis cinerea xylanase BcXyn11A elicits plant defenses. Front. Plant Sci. 2019, 10, 474 https://doi.org/10.3389/fpls.2019.00474.
The new title of the manuscript is: The study of some structural elements of a Pseudomonas fluorescens MF3 protein influencing its ability to induce resistance against fungi and other plant pathogens
Moreover, still some points need to be addressed as follows:
- It would be better to show the typical tobacco leaves together with Figure 3.
According to your recommendations, we added Fig. 4 showing tobacco leaves with typical necrotic lesions formed after their infection with TMV.
- The authors use the percent reduction of the lesion number on each leaf half to evaluate the reduction of inducing activity, however, the area of each lesion was different, so it is proper to use the area of all the lesions on each treatment.
The photographs of infected tobacco leaves (see Fig. 4) evidence that the areas of single necroses on different leaves may differ, but the difference between necroses developed on different halves of the same leaf is rather insignificant, so there is no need to use this index (lesion area) as an additional quantitative characteristics of the disease development level. At the same time, the numbers of necroses on the halves of the same leaf treated by MF3 and its mutant versions significantly differ. Due to this fact, the number of necroses is the main and common index illustrating the level of the disease development on necrotic-responding tobacco varieties (see highly cited basic paper: Holmes, F.O. Local lesions in tobacco mosaic. Botanical Gazette 1929, 87(1), 39‒55, https://doi.org/10.1086/333923). This method is also actively used today (see Nourinejhad Zarghani, S.; Ehlers, J.; Monavari, M.; von Bargen, S.; Hamacher, J.; Büttner, C.; Bandte, M. Applicability of different methods for quantifying virucidal efficacy using MENNO florades and tomato brown rugose fruit virus as an example. Plants 2023, 12, 894. https://doi.org/10.3390/plants12040894). Note also that the common methods for the evaluation of the disease development level by the lesion area use conventional international score scales developed for fungal and bacterial plant pathogens, but not for plant viruses (in this case, there is no such scoring scale).
- “2.1. Analysis of AC MF3 mutants inducing activity” here the subtitle should be 2.2; and “2.2. 3-D Model of MF3 and its analysis” here should be 2.3
Corrected
Reviewer 2 Report
Comments and Suggestions for Authors
The manuscript seems good, but needs correct
Introduction
I fhink, that Authors can mention about inoculation of soil and plants with fungi to increase plant immunity
For instances
Gleń-Karolczyk K., Boligłowa E., Antonkiewicz J. 2018. Organic fertilization shapes the biodiversity of fungal communities associated with potato dry rot. Applied Soil Ecology, 129, 43-51. https://doi.org/10.1016/j.apsoil.2018.04.012
Results and discussion
Please indicate the importance of this research in the context of food production
Please add conclusions and future for sciences
Author Response
We thank a reviewer for valuable comments intended to improve our manuscript.
Introduction
I fhink, that Authors can mention about inoculation of soil and plants with fungi to increase plant immunity
For instances
Gleń-Karolczyk K., Boligłowa E., Antonkiewicz J. 2018. Organic fertilization shapes the biodiversity of fungal communities associated with potato dry rot. Applied Soil Ecology, 129, 43-51. https://doi.org/10.1016/j.apsoil.2018.04.012
Done. We add some additional information about different eco-based methods to improve disease resistance of plants. All changes are indicated with yellow marking.
Results and discussion
Please indicate the importance of this research in the context of food production.
Done.
Please add conclusions and future for sciences
According to MDPI recommendations, the Conclusion section is required in the case of a large Discussion part; in other cases it is rather optional. That was the reason we did not use this section, but put concluding remarks at the end of Discussion. We enlarged this paragraph by adding the remarks concerning prospects of our further studies and a possible application of their results.
Round 2
Reviewer 1 Report
Comments and Suggestions for Authors
I would like to thank the authors for addressing my questions.
I have no more questions.
Reviewer 2 Report
Comments and Suggestions for Authors
In my opinion the manuscript has been corrected, improved according to reviewers.